# Thermophysical Properties of Laser Powder Bed Fused Ti-6Al-4V and AlSi10Mg Alloys Made with Varying Laser Parameters

**DOI:** 10.3390/ma16144920

**Published:** 2023-07-10

**Authors:** Stephen Akwaboa, Congyuan Zeng, Nigel Amoafo-Yeboah, Samuel Ibekwe, Patrick Mensah

**Affiliations:** 1Department of Mechanical Engineering, Southern University and A&M College, Baton Rouge, LA 70807, USA; stephen.akwaboa@sus.edu (S.A.); samuel_ibekwe@subr.edu (S.I.); patrick_mensah@subr.edu (P.M.); 2Department of Mechanical and Aerospace Engineering, North Carolina State University, Raleigh, NC 27695, USA; ntamoafo@ncsu.edu

**Keywords:** laser powder bed fusion, density, specific heat, thermal diffusivity, thermal conductivity

## Abstract

This study investigated the influence of diverse laser processing parameters on the thermophysical properties of Ti-6Al-4V and AlSi10Mg alloys manufactured via laser powder bed fusion. During fabrication, the laser power (50 W, 75 W, 100 W) and laser scanning speed (0.2 m/s, 0.4 m/s, 0.6 m/s) were adjusted while keeping other processing parameters constant. Besides laser processing parameters, this study also explored the impact of test temperatures on the thermophysical properties of the alloys. It was found that the thermophysical properties of L-PBF Ti-6Al-4V alloy samples were sensitive to laser processing parameters, while L-PBF AlSi10Mg alloy showed less sensitivity. In general, for the L-PBF Ti-6Al-4V alloy, as the laser power increased and laser scan speed decreased, both thermal diffusivity and conductivity increased. Both L-PBF Ti-6Al-4V and L-PBF AlSi10Mg alloys demonstrated similar dependence on test temperatures, with thermal diffusivity and conductivity increasing as the test temperature rose. The CALPHAD software Thermo-Calc (2023b), applied in Scheil Solidification Mode, was utilized to calculate the quantity of solution atoms, thus enhancing our understanding of observed thermal conductivity variations. A detailed analysis revealed how variations in laser processing parameters and test temperatures significantly influence the alloy’s resulting density, specific heat, thermal diffusivity, and thermal conductivity. This research not only highlights the importance of processing parameters but also enriches comprehension of the mechanisms influencing these effects in the domain of laser powder bed fusion.

## 1. Introduction

Additive manufacturing (AM), with its distinguished benefits, is increasingly influencing key sectors such as the automotive, aerospace, electronics, and biomedical industries [1,2,3]. These benefits include the ability to create parts with intricate shapes and efficiencies in terms of time and energy savings [4]. Among the multitude of AM techniques for metal part fabrication, laser powder bed fusion (L-PBF) has garnered significant attention. This method has facilitated the successful creation and comprehensive study of diverse metallic materials ranging from aluminum alloys [5], copper alloys [6], and titanium alloys [7], to nickel-based superalloys [8], stainless steels [9], and even high entropy alloys [10]. In a typical L-PBF process, a powder bed layer is evenly spread on top of the fabrication chamber, which is then selectively scanned with a laser, guided by the scanner system. Once the laser scanning is complete, another powder layer is placed on the previous one, and the process repeats. By building up layers, a three-dimensional part is finally fabricated. In this process, the laser spot size is typically around 50 μm, leading to a micro-sized melt pool. This minuscule melt pool allows for extremely rapid cooling rates of up to 10^6^ K/s, resulting in non-equilibrium structures within the L-PBF parts [11]. Additionally, parts fabricated through L-PBF are believed to contain significant amounts of residual stress, defects, and super-saturated solid solutions, all of which impact the properties of the parts, such as their mechanical, corrosion, and thermophysical properties [6,7,12].

Thermophysical properties are paramount when considering the application of these materials, particularly for thermal management applications. NASA researchers have recently carried out measurements on the thermophysical properties of various additively manufactured alloys for aerospace applications, such as Inconel 625, Inconel 718, Haynes 282, and stainless steel 316 L, primarily after heat treatments such as solution treatment and hot isostatic pressing (HIP) [2]. Given that thermophysical properties are susceptible to atomic-level defects, including point defects (solution atoms) and dislocations (residual stress) [13,14], investigating these properties in L-PBF parts is a significant undertaking. Furthermore, discerning the correlation between the structures and thermophysical properties holds substantial research value.

A few studies have delved into the thermal conductivity of L-PBF alloys, exploring the effects of factors such as sample building orientation, aging heat treatments, and the role of unique phase constituents. For instance, Zeng et al. [6] performed a study on the effects of sample building orientation and direct aging heat treatments on the thermal conductivity values of L-PBF Cu-Cr-Zr (C18150) alloy. They found that the sample building orientation had minimal influence on thermophysical properties, whereas aging heat treatments significantly affected them. Zeng et al. [13], Xie et al. [15], Yang et al. [16], Qi et al. [17], and Azizi et al. [18] all made similar contributions, analyzing different alloys under various conditions, with the goal of gaining deeper understanding into the thermophysical properties of L-PBF fabricated materials.

Despite these investigations, the impact of processing parameters on the thermophysical properties of L-PBF-produced metallic parts, and the underlying mechanisms governing these changes, remain unclear. In view of this, the current research fills a gap in understanding the influence of processing parameters on the thermophysical properties of metallic parts produced via L-PBF. By studying two different alloys, one with high thermal conductivity (AlSi10Mg) and another with low thermal conductivity (Ti-6Al-4V), this research investigates how these properties can be altered under different processing conditions. Although atomic defects are known to considerably affect thermophysical properties, they are not readily observable in experiments. To address this challenge, the CALPHAD method, employing Thermo-Calc software in Scheil Solidification Mode, was used to enhance the understanding of solid solutions in the as-fabricated L-PBF parts.

## 2. Materials and Methods

Commercial spherical shaped Ti-6Al-4V (20–50 μm, Concept-Laser GmbH, Lichtenfels, Germany) and AlSi10Mg (20–63 μm, Concept-Laser GmbH) alloy powders were used as the raw materials. The average compositions for the Ti-6Al-4V and AlSi10Mg alloys are shown in Table 1.

The 3D samples were made with a Concept-Laser GmbH Mlab-cusing-R system (Figure 1a) located in Louisiana State University (LSU), Baton Rouge, LA. The sample fabrication process was performed under argon atmosphere to minimize oxidation. To study the effects of processing parameters on thermal conductivity of the L-PBF parts, two parameters were selected to be varied, namely laser power and laser scan speed, while keeping hatch space (100 μm) and layer thickness (30 μm) constant. Laser power values of 50 W, 75 W, and 100 W, while laser scan speeds of 0.2 m/s, 0.4 m/s, and 0.6 m/s were chosen. Therefore, in this study, nine combinations of L-PBF samples were made for each alloy, which was shown in Table 2. Hereafter, for easy distinguishing, the samples were denoted according to the sample type and the fabrication parameters. For example, when the laser power and laser scanning speed were 50 W and 0.2 m/s, respectively, the L-PBF Ti-6Al-4V alloy and AlSi10Mg alloy were denoted as Ti-50-0.2 and Al-50-0.2, respectively (Table 2). In addition, the sample serial number in Table 2 will be utilized for x-axes in Figure 2 to easily distinguish the samples. For example, for Ti-6Al-4V alloy, “1” denotes Ti-50-0.2 sample, while for AlSi10Mg alloy, “1” represents Al-50-0.2 sample.

Temperature-dependent thermal conductivity ($K\left(T\right)$, W/(mK)) is obtained by multiplying temperature-dependent thermal diffusivity (α(T), mm^2^/s), specific heat ($C_{p}(T)$, J/(gK)), and density (ρ, g/cm^3^), which is shown in the following equation.
(1)$$K\left(T\right)=\alpha (T)\times C_{p}(T)\times \rho$$

Room-temperature density of the samples was determined using a gas displacement pycnometer (Micromeritics, AccuPyc II 1340, Norcross, GA, USA) (Figure 1b) with a disk-shaped sample (diameter of 8 mm and thickness of 5 mm). This equipment measures density by detecting pressure changes of helium within calibrated volumes using Archimedes Principle. Thermal diffusivity tests were performed using the Netzsch LFA 467 System, Selb, Germany (Figure 1c), with the sample diameter of 12.7 mm and thickness of 3 mm. For Ti-6Al-4V alloy, the thermophysical property test temperature range was 100–600 °C, while for AlSi10Mg alloy, the test temperature was from 100 °C to 400 °C. The specific heat of the test samples was determined using a reference sample (molybdenum, with known thermophysical properties). To ensure the repeatability and reliability of the test results, three samples were measured at each test temperature. Prior to testing, the sample surfaces were ground with SiC paper (600 mesh) and then sprayed with graphite to ensure identical flash-light energy absorption. A more detailed description of the thermal diffusivity and specific heat measurements can be found elsewhere [13]. To gain a better understanding of the phase structures and the corresponding phases compositions of the L-PBF Ti-6Al-4V and AlMg10Si alloys, CALPHAD method (Thermo-Calc software) was applied [19]. Ti-6Al-4V was calculated with the TCN8: Ni-Alloys v8.2 database, while AlSi10Mg was simulated with TCAL6: Al-Alloys v6.0 database. Scheil mode of solidification was utilized to simulate the fast L-PBF solidification process in this study.

## 3. Results and Discussion

Densities of the samples were measured at room temperature and the results are listed in Table 3. According to literature survey, the theoretical densities of Ti6Al4V and AlSi10Mg alloys are 4.43 g/cm^3^ [20], and 2.7 g/cm^3^ [21], respectively. It is intriguing and unexpected to find that some of the samples have density values exceeding their theoretical counterparts. This observation applies, for instance, to samples such as Ti-50-0.4, Ti-50-0.6, Ti-75-0.2, Ti-75-0.4, and Al-50-0.6. Typically, the densities of as-fabricated L-PBF parts are lower than the theoretical values due to the unavoidable presence of defects, such as vacancies and pores. Therefore, instances of relative densities exceeding 100% are most likely due to measurement uncertainties, a conclusion echoed by other researchers [22]. In this study, helium gas purging was used to determine the sample volumes. Helium gas would diffuse into the open defects (i.e., pores or cracks) of the L-PBF parts, reducing the measured sample volumes. As a result, the calculated densities of the samples appear larger than their actual values. Nonetheless, with the current density data, the preferred laser powers to achieve high densities in this study are 50 W and 75 W for the Ti-6Al-4V alloy and 100 W for the AlSi10Mg alloy.

Thermal conductivity was determined as the product of thermal diffusivity, specific heat, and density. The results of these thermophysical property tests for L-PBF Ti-6Al-4V alloy samples and AlSi10Mg alloy samples are displayed in Figure 2.

Specific heat shows different variation behaviors for L-PBF Ti-6Al-4V and AlSi10Mg alloys (Figure 2a,b). For the Ti-6Al-4V alloy under each laser power, an increase in laser scan speed initially decreases the specific heat values (from 0.2 m/s to 0.4 m/s), then causes it to rise (from 0.4 m/s to 0.6 m/s). Concurrently, under each sample condition (with the same laser power and laser scan speed), the specific heat generally increases as the test temperature rises (Figure 2a). Various factors can explain the change in specific heat as a function of laser scan speed. As is widely accepted, higher laser scan speeds result in faster cooling rates and higher thermal gradients, which lead to finer microstructures and phase structures, and vice versa [23,24]. The cooling rate and thermal gradients influence the formation of different-sized grains and phases with varying size, composition, and distribution [25]. Smaller grains yield more grain boundaries, disrupting heat flow and effectively increasing specific heat values [26,27]. Uniformly distributed secondary phases within the sample produce different behaviors of specific heat compared to when the same phases segregate into different regions. Moreover, phases with different compositions also lead to varying specific heat [28]. Consequently, the observed decrease and subsequent increase of specific heat as a function of laser scan speed can be attributed to the combined effects of the above factors, namely (i) changes in grain size and (ii) variations in secondary phases (including their size, distribution, and compositions) induced by changes in the cooling rate and thermal gradients.

When compared to the L-PBF Ti-6Al-4V alloy scenario, there are no discernible systematic variation patterns for the AlSi10Mg case with respect to specific heat values (Figure 2b). The notable exception to this is that the specific values for the Al-50-0.6 sample are significantly lower than those of other L-PBF samples fabricated using different laser parameters. It’s important to consider that the Al-50-0.6 sample was fabricated with the lowest energy density. This likely resulted in the quickest cooling rate, leading to the formation of unique non-equilibrium phases characterized by lower specific heat values.

In general, the specific heat of both L-PBF Ti-6Al-4V (Figure 2a) and AlSi10Mg (Figure 2b) alloys rises as the test temperature increases. Similar trends have been observed in the majority of other alloys [2]. This behavior can be attributed to several factors. First, as temperatures climb, atoms vibrate more intensely. For the temperature to increase further, more energy is required to facilitate even stronger atomic vibrations. This, in turn, results in higher specific heat values at elevated temperatures [29]. Second, in metallic materials, as the temperature rises, electrons are excited to higher energy states. These additional excitations necessitate more energy, contributing to higher specific heat values [30]. Third, materials tend to expand as the temperature increases. The energy absorbed during this thermal expansion process also boosts the specific heat values [31]. Fourth, in crystalline materials, heat is primarily transported by phonons, which are quantized modes of vibration occurring at the atomic scale. At high temperatures, these phonons interact more vigorously with each other, which further increases the specific heat [32]. In summary, the general increase in specific heat with rising temperatures can be attributed to: (i) enhanced atomic vibrations; (ii) more electrons being excited to higher energy states; (iii) thermal expansion; and (iv) stronger interactions between phonons at elevated temperatures.

For both the Ti-6Al-4V alloy and the AlSi10Mg alloy, the thermal diffusivity and thermal conductivity test results exhibit similar patterns of variation, implying that thermal diffusivity significantly impacts thermal conductivity (Figure 2c–f). Similar to the case with specific heat, both thermal diffusivity and thermal conductivity are influenced by processing parameters and test temperatures. Particularly for L-PBF Ti-6Al-4V alloy samples, with a fixed laser power, thermal diffusivity and conductivity tend to decrease as the laser scan speed increases. Conversely, at a constant laser scan speed, thermal diffusivity and conductivity rise with an increase in laser power. Additionally for L-PBF alloy samples, thermal diffusivity and conductivity increase as test temperatures rise, except for the results observed at 400 °C for L-PBF AlSi10Mg alloy samples.

The dependency of thermal diffusivity and conductivity on processing parameters can be elucidated as follows: with a constant laser power, an increase in laser scan speed results in a higher cooling rate, while with a fixed laser scanning speed, a lower laser power also leads to a quicker cooling rate [33]. The rapid cooling rate positions the L-PBF process in the realm of solution heat treatment processes. In the solution treatment processes, alloys are heated and maintained at a specific temperature (not exceeding the melting point). At this temperature, solute atoms dissolve completely into the solvent atoms (primary metal atoms). Then, the alloy is rapidly cooled to room temperature through a quenching process, usually involving immersion in water or oil. This rapid cooling solidifies the solute atoms, preventing them from precipitating out of the solution. During the L-PBF process, metal powders are first melted by the laser energy source and then solidify extremely rapidly, at rates of up to 10^6^ K/s [11]. This intense cooling rate preserves the distribution of the solute atoms within the matrix atoms, forming a super-saturated solid solution.

Thermo-Calc was employed to calculate phase structures of Ti-6Al-4V and AlSi10Mg alloys under both equilibrium and non-equilibrium (fast cooling) conditions. Figure 3 displays the proportions of all phases as a function of temperature in equilibrium states (adequate diffusion) for Ti-6Al-4V and AlSi10Mg alloys. As depicted in the images, the stable phases, and their respective compositions at room temperature (RT) are provided in Table 4. Multiple phases are present in both AlSi10Mg and Ti-6Al-4V alloys under equilibrium solidification conditions. In such cases, the primary phases for AlSi10Mg and Ti-6Al-4V alloys are FCC_A1 and HCP_A3 phases, respectively. Detailed examination of the phase composition data in Table 4 reveals that the FCC_Al phase in AlSi10Mg alloys is theoretically pure Al, while the composition of the HCP_A3 phase in Ti-6Al-4V alloy is nearly Ti-6.25Al.

To obtain a better understanding of the effects of cooling rate on the phase structures and individual phase compositions, the Scheil mode of solidification was utilized in Thermo-Calc. This mode assumes that no diffusion occurs in the solid state, while infinite diffusion occurs in the liquid phase [34]. The simulation results for AlSi10Mg and Ti-6Al-4V using the Scheil mode are demonstrated in the Figure 4. When compared to the Figure 3 illustrating solidification under equilibrium conditions, distinct phases emerge under rapid cooling states. Specifically, the Al18Fe2Mg7Si10 phase appears in the AlSi10Mg alloy instead of the Al9Fe2Si2 phase, and only the BCC_B2 phase appears rather than the HCP_A3 + BCC_B2 binary phases present during equilibrium cooling for Ti-6Al-4V alloy. The cooling rate evidently exerts a substantial influence on the phase structures of the alloys. The phase compositions of the alloys were also carefully investigated, with the results listed in Table 4 and shown in the Figure 5. By comparing these results, it is evident that the principal phases of the alloys under equilibrium conditions contain significantly lower quantity of solute atoms. Specifically, for the AlSi10Mg alloy, it contains 100% Al under equilibrium conditions in the FCC_Al phase (Table 4), while it possesses approximately 2 wt.% Mg + Si solute atoms under fast-cooling state (Figure 5). For Ti-6Al-4V, there are only ~6.25 wt.% Al solute atoms in the major HCP_A3 phase under equilibrium states (Table 4), while almost all of the 10 wt.% Al + Si atoms remain in the Ti matrix as a single BCC_B2 phase under fast cooling conditions (Figure 5). Consequently, due to rapid cooling, a higher content of solute atoms is present in L-PBF samples.

In alloys, solid solute atoms are potent scatterers of electrons. This electron scattering reduces the alloys’ electrical conductivity [35]. The Wiedemann–Franz law [36] connects electrical conductivity and electronic thermal conductivity, as described in Equation (2):(2)$$\lambda_{e}=L\sigma T$$
where, $\lambda_{e}$ is the electronic thermal conductivity, L is the Lorentz number, σ is the electrical conductivity, and *T* is temperature in Kelvins. The scattering by solute atoms in the solid solution, which reduces electrical conductivity, results in a corresponding decrease in thermal conductivity contributed by electrons. Simultaneously, the presence of solute atoms disrupts the regularity of the crystal lattice, impeding the efficient transfer of thermal energy by phonons and, in turn, lowering the lattice thermal conductivity of the alloy [37]. The overall thermal conductivity of the alloy is the sum of the electronic and lattice thermal conductivities. Hence, the presence of solute atoms can reduce the alloy’s total thermal conductivity. Prior research has shown that a higher content of solute atoms tends to yield lower overall thermal conductivity [13]. Regarding L-PBF Ti-6Al-4V alloy samples with constant laser power, such as Ti-50-0.2, Ti-50-0.4, and Ti-50-0.6 samples, an increase in laser scan speed reduces the size of the melt pools of laser scan tracks, resulting in a faster cooling rate. This accelerated cooling leaves less time for the solute atoms in the Ti-50-0.6 sample to diffuse and form precipitates. Consequently, compared to the Ti-50-0.2 and Ti-50-0.4 samples, the Ti-50-0.6 sample contains a higher concentration of solute atoms, which leads to lower thermal diffusivity/conductivity. This rationale also applies to the other two groups of samples, which maintain a constant laser power of 75 W and 100 W. When the laser scan speed is constant, as in the Ti-50-0.2, Ti-75-0.2, and Ti-100-0.2 alloy samples, higher laser power leads to a larger melt pool size, resulting in a slower cooling rate, allowing more time for the solute atoms to diffuse and precipitate. As a result, the Ti-100-0.2 sample contains the smallest number of solute atoms and therefore exhibits the highest thermal diffusivity/conductivity within this group. This explanation also holds true for the other two sample groups, which maintain constant laser scan speeds of 0.4 m/s and 0.6 m/s.

In addition, a higher cooling rate generally results in a finer grain size [38]. As per previous studies, the grain size post the L-PBF process typically falls within the microscale [13,39]. However, to substantially influence thermal conductivity, the grain size must be reduced further to the nanoscale, or more precisely, to a size comparable to the mean free path of heat carriers such as electrons and phonons, which exists at the atomic level [40,41,42,43]. Consequently, grain boundary (grain size) scattering is likely not the primary factor affecting the thermal diffusivity/conductivity of the L-PBF Ti-6Al-4V alloy.

While solute atoms significantly influence the thermal diffusivity/conductivity of the L-PBF Ti-6Al-4V alloy, this does not seem to be the case for the L-PBF AlSi10Mg alloy in this study (Figure 2d,f). No distinct variation in thermal diffusivity/conductivity is observed for the L-PBF AlSi10Mg alloy, except that the thermal diffusivity/conductivity of the samples created with a constant laser power (50 W) is lower than those fabricated with 75 W and 100 W. This finding suggests that to achieve stable thermal diffusivity/conductivity for AlSi10Mg alloy, the laser power should be either 75 W or 100 W in this study. When the laser power surpasses 75 W, no notable variation is discernible for the samples, including Al-75-0.2, Al-75-0.4, Al-75-0.6, Al-100-0.2, Al-100-0.4, and Al-100-0.6 samples. This implies that the combined effect of defects (especially point defects such as solute atoms) and grain size is similar across these six samples, resulting in close thermal diffusivity/conductivity values.

Interestingly, it is important to note that the thermal diffusivity/conductivity of both L-PBF Ti-6Al-4V and AlSi10Mg alloy samples generally increases with rising test temperature (Figure 2c–f). This observation is intriguing as this trend contrasts with the behavior of pure metals, such as copper, aluminum, and titanium [6,44,45]. The thermal diffusivity/conductivity of these pure metals decreases with increasing test temperature (over the temperature range explored in this study). The explanation for these contradictory behaviors in thermal diffusivity/conductivity (increasing with rising test temperature for alloys, while decreasing with increased test temperature for pure metals) is discussed below.

The variation in temperature-dependent thermal conductivities of pure metals and alloys, from room temperature to relatively high temperatures (600 °C for the Ti-6Al-4V alloy and 400 °C for the AlSi10Mg alloy), can primarily be attributed to differences in their atomic and electronic structures, which ultimately influence their heat transfer capabilities. Specifically, in pure metals, heat conduction is mainly governed by the scattering of phonons and electrons [46]. Pure metals often demonstrate high thermal conductivity due to their atoms being identical and systematically arranged in a crystalline lattice and owing to their possession of a ‘sea’ of freely moving electrons. At lower temperatures, the thermal diffusivity/conductivity of pure metals is high due to fewer lattice vibrations that scatter the electrons. However, as the test temperature increases, the atoms vibrate more intensely, leading to enhanced phonon-electron scattering. This impedes the flow of heat and boosts thermal resistance. Therefore, throughout the currently tested temperature range, thermal diffusivity/conductivity decreases with an increase in test temperature [47].

In contrast to pure metals, which contain only one type of atom, alloys are mixtures of two or more elements, with at least one element being a metal. The presence of different types of atoms in an alloy gives rise to a more disordered lattice structure, increasing the scattering of phonons and electrons (termed impurity scattering) [48,49]. At lower temperatures, the impact of impurity scattering dominates, usually resulting in alloys having lower thermal conductivity than their constituent pure metals. Nevertheless, as the test temperature increases, heightened phonon-phonon scattering leads to increased thermal conductivity since this effect supersedes the impurity scattering caused by lattice disorder. Consequently, generally, the thermal conductivity of alloys begins to increase with temperature. However, beyond a certain temperature, amplified phonon-phonon scattering would start to hinder thermal flow more than it facilitates it, leading to a decrease in thermal conductivity [47]. This perfectly explains the observed drop in thermal diffusivity at 400 °C for L-PBF AlSi10Mg alloy.

## 4. Conclusions

This study investigated the influence of laser fabrication parameters and test temperatures on the thermophysical properties of Ti-6Al-4V and AlSi10Mg alloys fabricated using laser powder bed fusion (L-PBF). The investigation utilized both experimental and simulation studies, together with literature review. The primary conclusions drawn from the study are as follows:(1)As the test temperature rises, the specific heat values increase due to several factors, including enhanced atomic vibrations, excitation of more electrons to higher energy states, thermal expansion, and stronger phonon interactions at elevated temperatures.(2)The Ti-6Al-4V alloy exhibits low thermal conductivity, while the AlSi10Mg alloy demonstrates high thermal conductivity. The response of these alloys to changes in laser processing parameters varies. Thermophysical properties of L-PBF Ti-6Al-4V parts are significantly influenced by laser processing parameters. In contrast, the L-PBF AlSi10Mg alloy in this study does not demonstrate a clear sensitivity to variations in laser processing parameters. Specifically, at 100 °C, with the variation of laser processing parameters used in this study, the thermal conductivity of L-PBF Ti-6Al-4V alloy ranges from 2.6 to 8.5 W/(mK). This phenomenon is due to the fact that a higher cooling rate leads to an increase in the quantity of solute atoms within the L-PBF parts, as confirmed by Thermo-Calc simulation, which results in a reduction of thermal diffusivity and conductivity.(3)Furthermore, for both L-PBF Ti-6Al-4V and AlSi10Mg alloys, thermal conductivity increases with test temperature; for example, it rises from 8.5 to 16.4 W/(mK) and from 118.2 to 172.0 W/(mK) for L-PBF Ti-6Al-4V and AlSi10Mg alloys fabricated with a laser power of 100 W, and a laser scan speed of 200 mm/s, respectively. This trend can be attributed to the competition between impurity scattering and phonon-phonon scattering.

## Figures and Tables

**Figure 1 materials-16-04920-f001:**
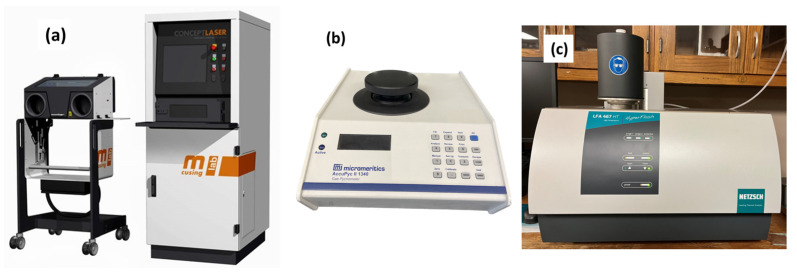
Images showing the experimental setups, (**a**) laser powder bed fusion 3D printer from Concept-Laser, (**b**) pycnometer for density measurement, (**c**) Netzsch LFA 467 thermal diffusivity tester.

**Figure 2 materials-16-04920-f002:**
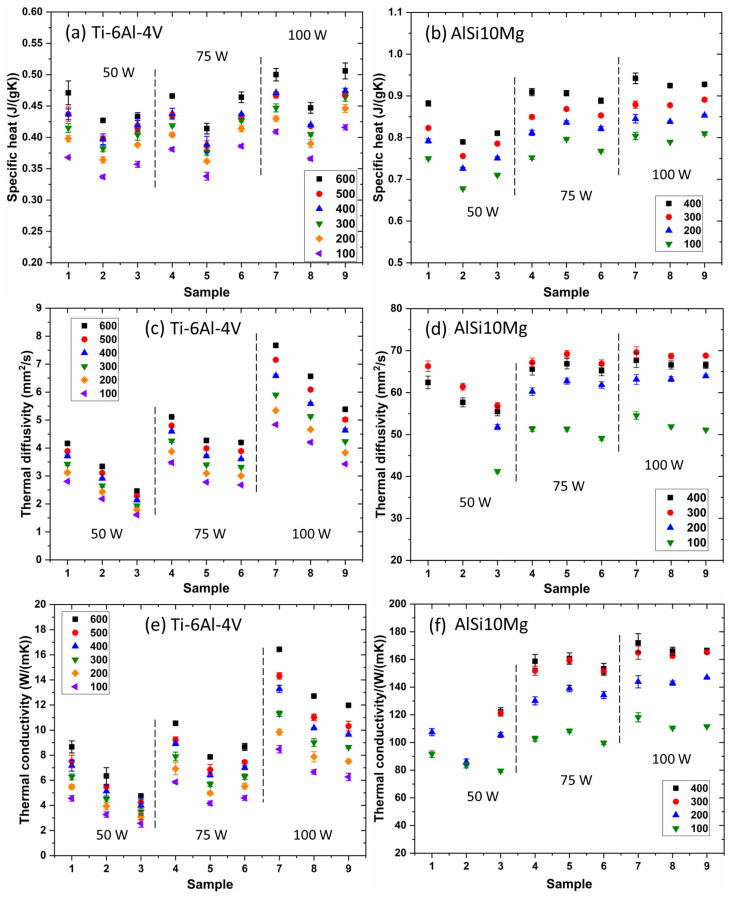
Image showing the thermophysical property test results of L-PBF Ti-6Al-4V and AlSi10Mg alloy samples from 100 °C to 600 °C and 400 °C, respectively. Specifically, this figure shows results for (**a**) specific heat, (**c**) thermal diffusivity, and (**e**) thermal conductivity for the L-PBF Ti-6Al-4V alloy samples, and (**b**) specific heat, (**d**) thermal diffusivity, and (**f**) thermal conductivity for the L-PBF AlSi10Mg alloys. In this figure, the number “100”, “200”, “300”, “400”, “500”, and “600” denote the test temperatures in the unit of °C.

**Figure 3 materials-16-04920-f003:**
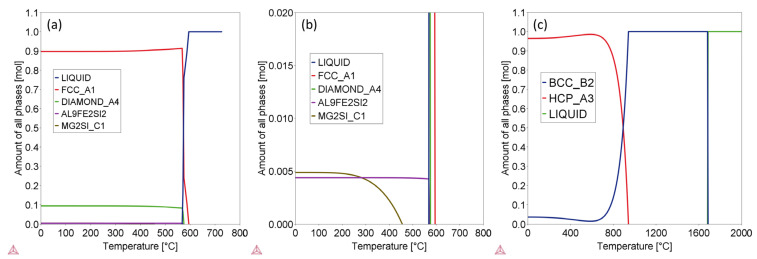
Images showing the amounts of all phases in the alloys as a function of temperature at equilibrium state. (**a**) Full phase content range for AlSi10Mg; (**b**) narrow phase content range for AlSi10Mg; (**c**) full phase content range for Ti-6Al-4V.

**Figure 4 materials-16-04920-f004:**
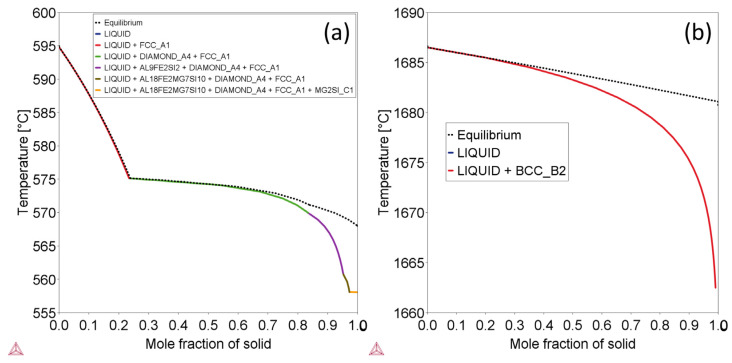
Images showing the phase changes during solidification process for AlSi10Mg alloy (**a**) and Ti-6Al-4V alloy (**b**) under non-equilibrium states.

**Figure 5 materials-16-04920-f005:**
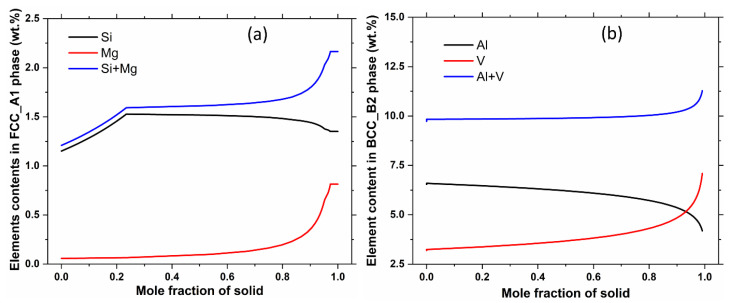
Images indicating the mass contents of solution atoms in the main phases in (**a**) AlSi10Mg, and (**b**) Ti-6Al-4V alloys under Scheil model of solidification.

**Table 1 materials-16-04920-t001:** Average compositions for the Ti-6Al-4V and AlSi10Mg alloy powders.

Alloy	Element Content (wt.%)					
	Ti	Al	V	Si	Mg	Fe
Ti-6Al-4V	Bal.	6.02	3.98	/	/	/
AlSi10Mg	/	Bal.	/	9.92	0.291	0.137

**Table 2 materials-16-04920-t002:** Sample processing design and denotation for the samples.

Samples	Laser Power (W)	Scanning Speed (m/s)	Sample Denotation for Ti-6Al-4V Alloy	Sample Denotation for AlSi10Mg Alloy
1	50	0.2	Ti-50-0.2	Al-50-0.2
2	50	0.4	Ti-50-0.4	Al-50-0.4
3	50	0.6	Ti-50-0.6	Al-50-0.6
4	75	0.2	Ti-75-0.2	Al-75-0.2
5	75	0.4	Ti-75-0.4	Al-75-0.4
6	75	0.6	Ti-75-0.6	Al-75-0.6
7	100	0.2	Ti-100-0.2	Al-100-0.2
8	100	0.4	Ti-100-0.4	Al-100-0.4
9	100	0.6	Ti-100-0.6	Al-100-0.6

**Table 3 materials-16-04920-t003:** Measured densities and the corresponding relative density values for the L-PBF Ti-6Al-4V and AlSi10Mg alloy samples.

Sample	Ti-6Al-4V		AlSi10Mg	
	Density (g/cm^3^)	Relative Density * (%)	Density (g/cm^3^)	Relative Density * (%)
50-0.2	4.417 ± 0.006	99.8	2.620 ± 0.003	97.0
50-0.4	4.448 ± 0.016	100.5	2.681 ± 0.007	99.3
50-0.6	4.465 ± 0.008	100.9	2.715 ± 0.005	100.4
75-0.2	4.425 ± 0.012	99.8	2.661 ± 0.008	98.5
75-0.4	4.444 ± 0.002	100.2	2.651 ± 0.004	98.1
75-0.6	4.444 ± 0.017	100.2	2.644 ± 0.006	97.8
100-0.2	4.286 ± 0.016	96.8	2.696 ± 0.005	100.0
100-0.4	4.331 ± 0.007	97.7	2.697 ± 0.003	100.0
100-0.6	4.394 ± 0.006	99.1	2.695 ± 0.005	99.6

* The reference density for the relative density is the theoretical density.

**Table 4 materials-16-04920-t004:** Stable phases, the phase mass contents and individual phase compositions of the Ti-6Al-4V alloy and AlSi10Mg alloy under equilibrium states.

Alloy	Stable Phases at RT	Phase Content (wt.%)	Phase Composition					
			Ti	Al	V	Si	Mg	Fe
Ti-6Al-4V	HCP_A3	96.01	0.9373	0.0625	0.0002	/	/	/
	BCC_B2	3.99	0.0028	0.0002	0.9971	/	/	/
AlSi10Mg	AL9FE2SI2	0.51	/	0.6411	/	0.0892	/	0.2697
	Diamond_A4	9.71	/	/	/	1.0000	/	/
	FCC_A1	89.33	/	1.0000	/	/	/	/
	MG2SI_C1	0.46	/	/	/	0.3662	0.6338	/

## Data Availability

The raw data supporting the conclusions of this article will be made available by the authors, without undue reservation, to any qualified researcher. For additional information on the datasets, please contact the corresponding author.

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
