# Peer review of "Thermophysical Properties of Laser Powder Bed Fused Ti-6Al-4V and AlSi10Mg Alloys Made with Varying Laser Parameters"

_materials, 2023, doi:10.3390/ma16144920_

Round 1
Reviewer 1 Report
This manuscript investigates the variations in thermophysical properties of Ti-6Al-4V and AlSi10Mg alloy parts fabricated through laser bed fusion using an array of processing parameters. The paper is well-organized, and the title is practical and attractive. Still, it is associated with two basic challenges of innovation and experimental design to choose the same parameters for two substances. In addition, the following should also be considered.
What is the purpose of research? These alloys are the most widely used in SLM, and the effect of printing parameters has been investigated more comprehensively in articles. What is the difference between your work and previous articles? The abstract should be written better and needs major revisions. The purpose of research and innovation should be clearly stated. Also, the performed tests should be presented first, and then the results should be presented quantitatively and qualitatively. Referencing the articles is not acceptable in line 26. The use of general sentences with more than five references can be seen in all parts of the introduction. On the other hand, appropriate references were not used to analyze the results. The introduction is very general. Although the introduction is long, it is written superficially in some paragraphs. Also, in the end, a suitable summary of the importance of the present issue should be provided. Use the following resources to deepen the introduction. The high temperature flow behavior of additively manufactured Inconel 625 superalloy. Review of selective laser melting of magnesium alloys: Advantages, microstructure and mechanical characterizations, defects, challenges, and applications. Effects of post-weld heat treatment on the microstructure and mechanical properties of laser-welded NiTi/304SS joint with Ni filler. Laser powder bed fusion of Alumina/Fe–Ni ceramic matrix particulate composites impregnated with a polymeric resin. Figures 1 and 2 as well as 3 and 4, can be combined in two parts in one figure. How has the reproducibility of these results been checked? Specify the number of experimental tests conducted. Errorbar should be added to all numerical results. Why are the printing parameters considered the same for both alloys? Why is the effect of the material on the printing parameters not considered? On what basis are these parameters selected? Has the manufacturer's proposal been used or previous sources? Other printing parameters can also be provided.
No comment.
Author Response
The authors really appreciate the constructive comments of the reviewer, which is of great help to improve the quality of this paper. The responses of the authors to the comments can be found in the attachment, which were highlighted in blue for easy distinguishing.

Reviewer 2 Report
The manuscript examines the thermophysical properties of two distinct alloys created through LPBF, while varying laser power and scanning speed. In my opinion, the obtained results have originality and the manuscript was well-written. Nevertheless, there are certain aspects, particularly concerning the presentation of the abstract, results and the conclusion, that require revision prior to accepting the paper.
1. The Abstract section lacks the presentation of results pertaining to the influence of test temperature on the thermophysical properties of the manufactured specimens, which were extensively discussed throughout the manuscript.
2. The rationale behind selecting 50 W and 75 W for Ti-6Al-4V, and 100 W for AlSi10Mg as the laser powers resulting in higher density remains unclear. The discussion regarding these results is missing.
3. The positioning of Figures 1-6 in consecutive order makes it inconvenient for readers to simultaneously refer to the figures and accompanying text. Additionally, it would be beneficial to place the thermophysical property figures for Ti-6Al-4V and AlSi10Mg side by side for easier comparison.
4. While the manuscript mentions the possible reasons for the observed trend in specific heat values for the Ti-6Al-4V alloy—initial decrease with increasing laser scanning speed (0.2 m/s to 0.4 m/s), followed by an increase (0.4 m/s to 0.6 m/s)—it does not provide a clear explanation for why this initial decrease and subsequent increase occur.
5. In a study by Kim MS (2021) titled "Effects of processing parameters of selective laser melting process on thermal conductivity of AlSi10Mg alloy, (Materials 14:2410)" it was observed that the thermal diffusivity of LPBF processed AlSi10Mg alloy decreases with increasing test temperature. The discrepancy between the results of this study and the study by Kim MS should be addressed.
6. The conclusion section needs revision to account for the discrepancies observed in the results of the AlSi10Mg and Ti-6Al-4V alloys.
Author Response
The authors would like to express their gratitude to the reviewer’s valuable comments, which is of great importance to improve the quality of this paper. The responses of the authors to the comments can be found in the attachment, which were highlighted in blue for easy distinguishing.

Reviewer 3 Report
In this article, the authors presented a study on the effect of laser fabrication parameters on the thermo-physical properties of Ti-6Al-4V and AlSi10Mg alloys that were fabricated using L-PBF. The authors described the results from systematic investigation and discussed them. Although the paper has merits, it could be improved. Following are my comments.
To evaluate the effects of processing parameters on thermal conductivity of alloys, the authors tested laser power values of 50 W, 75 W, and 100 W, and laser scanning speeds of 0.2 m/s, 0.4 m/s, and 0.6 m/s. Why did the authors choose these range for the parameters? Why not choose 25 or 150 W, for example?
Could the authors give the unity of the thermal conductivity, thermal diffusivity, specific heat, and density, for the equation (1)?
What could be the measurements uncertainties causing a relative density exceeding 100%?
The data show on Figures 1-3 is a little bit confusing. In the legend of the figures, it is not clear that the values 100, 200, 300, 400, 500 and 600 are the test temperature. The graph could be better explained and discussed. Also, authors should include the unity of the value in the legend.
The graphs are confusing. The colors used for the temperatures, samples, etc. are not standardized. Authors should improve the Figures 1 to 6.
Authors identify the Figures 2 and 5 as the ones with the graph for specific heat for both samples in the paragraph of page 5 but not on the paragraph of page 4, the first one discussing the results for specific heat.
Why the thermal diffusivity and conductivity has a different behavior for the AlSi10Mg alloy at 400 ℃?
Authors could discuss deeper the thermal diffusivity and conductivity. And, also, reorder the graphs in Figures 1 to 6 because the discussion is focus on Figures 2 and 5.
Why would a phonon-phonon scattering increase the thermal conductivity if this effect is, as pointed by the author in the preceding paragraph, responsible for decreasing heat conductance? Also, is not the increase in the thermal conductivity in alloys with rising temperature predicted by equation (2)?
Author Response
The authors really appreciate the reviewer’s careful and constructive comments, which is of great importance to improve the overall quality of this paper. The authors would like to express their gratitude to the reviewer’s valuable comments, which is of great importance to improve the quality of this paper.

Reviewer 4 Report
Pls revise your manuscript based on the following points:
-abstract needs revision and extension. Pls properly brief all the necessary result/outcome points both qualitative and quantitative.
-In Conclusion also, you should mention quantitative results.
-It is recommended to provide the actual pic of experimental setup/machine used, in section 2 materials and method.
-Results should be supported with SEM images.
Rest is fine.
Author Response
The authors really appreciate the reviewer’s careful and constructive comments, which is of great importance to improve the overall quality of this paper. The responses of the authors to the comments can be found in the attachment.

Round 2
Reviewer 3 Report
Although authors have made the changes and the manuscript could been accepted in the preent form, the data show in Table 3 is incorrect. The mean value and the uncertainties are expressed incorrectly regarding the significant algarisms. Authors should correct it.
Author Response
Dear Reviewer,
Thank you for your attentive review and your comment concerning Table 3. We apologize for the discrepancy you've noticed in the mean value and uncertainties in relation to significant figures.
We have revised the Table 3 accordingly. Thank you again for your valuable comments that help to significantly improve the quality of this paper!
Best regards,
Congyuan Zeng